# Cooperative Energy-Efficient Routing Protocol for Underwater Wireless Sensor Networks

**DOI:** 10.3390/s22186945

**Published:** 2022-09-14

**Authors:** Irfan Ahmad, Taj Rahman, Asim Zeb, Inayat Khan, Mohamed Tahar Ben Othman, Habib Hamam

**Affiliations:** 1Department of Physical & Numerical Science, Qurtuba University of Science & Information Technology, Peshawar 25000, Pakistan; 3Department of Computer Science, Abbottabad University of Science and Technology, Abbottabad 22500, Pakistan; 4Department of Computer Science, University of Buner, Buner 19290, Pakistan; 5Department of Computer Science, College of Computer, Qassim University, Buraydah 51452, Saudi Arabia; 6Faculty of Engineering, Uni de Moncton, Moncton, NB E1A 3E9, Canada; 7Spectrum of Knowledge Production & Skills Development, Sfax 3027, Tunisia; 8International Institute of Technology and Management, Libreville BP1989, Gabon; 9Department of Electrical and Electronic Engineering Science, School of Electrical Engineering, University of Johannesburg, Johannesburg 2006, South Africa

**Keywords:** UWSNs, energy-efficient routing, CEER, PDR, cooperative routing, sink node

## Abstract

Underwater wireless sensor networks (UWSNs) contain sensor nodes that sense the data and then transfer them to the sink node or base station. Sensor nodes are operationalized through limited-power batteries. Therefore, improvement in energy consumption becomes critical in UWSNs. Data forwarding through the nearest sensor node to the sink or base station reduces the network’s reliability and stability because it creates a hotspot and drains the energy early. In this paper, we propose the cooperative energy-efficient routing (CEER) protocol to increase the network lifetime and acquire a reliable network. We use the sink mobility scheme to reduce energy consumption by eliminating the hotspot issue. We have divided the area into multiple sections for better deployment and deployed the sink nodes in each area. Sensor nodes generate the data and send it to the sink nodes to reduce energy consumption. We have also used the cooperative technique to achieve reliability in the network. Based on simulation results, the proposed scheme performed better than existing routing protocols in terms of packet delivery ratio (PDR), energy consumption, transmission loss, and end-to-end delay.

## 1. Background

UWSNs have become a significant area for military surveillance, environmental monitoring, and hidden resources. In a typical UWSN, sensor nodes are connected to sink nodes, surface stations, and other nodes in this environment [1]. Acoustic signals are used instead of radio signals in UWSNs to transfer data from the source to the destination because the saltwater firmly interrupts the radio signals. A decentralized UWSN can provide a low-cost solution for rapidly deploying sensors to measure the parameters that may harm the marine environment. Such as defining the quantity and location of an oil or fuel leak, evaluating the surface area and direction of toxic algae banks due to the effect of marine currents, etc. [2].

Along with monitoring the specific jobs in deep and shallow water, UWSN communications are interrupted by undesirable issues such as limited bandwidth, high propagation delay, bit error rate, and high energy consumption [2]. Among the most significant unwanted effects in UWSNs is node dying because of inefficient energy use, typically caused by improper route selection. An efficient and energy consumption routing protocol is required concerning the mentioned issues and challenges. The primary factor determining a routing protocol is selecting the relaying node. This node selection mechanism depends on several factors, such as distance, hop count, and residual energy.

Nodes in UWSNs have limited battery power, and battery replacement is nearly impossible in such a restricted environment. A node’s limited-power battery must be considered while developing a routing protocol. The proper deployment of sinks, sensor devices, and other equipment is critical in increasing the network’s life [3]. Furthermore, the network topology also plays an essential role in reducing energy consumption in UWSNs. It means less energy will be used if the network topology is well designed and managed; otherwise, more energy will be wasted, and the nodes will die early. The structure of UWSNs is shown in Figure 1.

The designed protocols have mainly focused on improving the lifetime of the network. As previously mentioned, replacement of the sensor node battery and recharge are extremely challenging in sensor devices, so routing protocols must address energy consumption balance. There are two ways to transfer the data from the source to the destination: noncooperative or using a single node for transferring the data. The second one is cooperative communication or using the relay nodes instead of a single node. Cooperative communication is the best way to provide reliable communication between the nodes and minimize data failures in these networks. Cooperative communication transfers data via numerous routes to increase the possibility of receiving it correctly at its final destination [4,5].

Cooperative methods are also divided into incremental relaying nodes and fixed relaying nodes. The fixed relying method refers to data improvement by the relay for data reliability. During incremental collaboration, delivery occurs when the receiver requests it. The relay either boosts or decodes data before sending it to the forwarder/receiver. The cooperative methods improve data receiving and reduce packet loss rate. However, significant energy and consumption of time make them difficult or impossible. It is more difficult in the case of acoustic waves due to limited energy resources and communication speed.

In the noncooperative method, data delivery through a single link uses less energy and time than cooperative approaches. However, these techniques are unreliable and have a higher probability of dropping data packets. Noise sources damage data in noncooperative methods and have a more significant impact on data loss. Retransmission is not appropriate for data received over the same noisy link. Additionally, many antennas are expensive and impractical in the underwater environment [6]. Cooperation is the optimal strategy for ensuring reliable and effective communication in UAWSNs.

### 1.1. Motivation

The random deployment of UWSNs nodes, the difficulty of charging these nodes, minimizing power consumption, propagation delay, and bandwidth issues must all be addressed. Data forwarding through multiple nodes from source to destination is better to solve the network reliability issues. Still, the latency will be high due to various nodes in the data processing. The motivation of this paper is from [7], where the authors focused on reducing energy consumption, increasing network lifetime, and enhancing the packet delivery ratio. Data is transferred on demand to avoid voids and trapped nodes in the data participating process. Due to the abovementioned issues, we proposed the cooperative energy-efficient routing protocol (CEER) to increase the network’s lifetime and reliability.

### 1.2. Contribution

The proposed routing scheme is designed to reduce energy consumption and latency issues. Therefore, the proposed method uses the sink mobility technique to transfer data to the sink node. This will minimize energy consumption because of the direct transmission of data. When the sink is in communication range, the nodes will transfer the data directly to the sink.Data transferring through a single link does not ensure the reliability of data. Therefore, the proposed method uses the cooperative data forwarding scheme to reduce the end-to-end delay and increase the network’s reliability.The proposed routing scheme performance is evaluated through the MATLAB simulation tool, which shows the improvement in energy consumption, end-to-end delay, PDR, and transmission loss.

### 1.3. Paper Organization

The rest of the paper is organized as follows. The main work of the existing methods is discussed in Section 2. The proposed CEER routing protocol with necessary details is discussed in Section 3. Section 4 is the MATLAB simulation of both the proposed and existing two routing protocols are discussed briefly. The last section discusses the conclusion of the paper.

## 2. Literature Review

The two most critical issues of UWSNs, such as data transferring in the harsh environment through the noisy links and sensor node energy consumption, are addressed in [8]. The authors proposed two routing protocols in the mentioned work. The first is energy-effective and reliable delivery (EERD), and the second is cooperative energy-effective and reliable delivery (CoEERD). The EERD incorporates the sensor node’s energy consumption parameter, and the CoEERD addresses the second issue, transferring data through the noisy links. In EERD, data are transferred from source to destination through a single link. The forwarder nodes are selected based on the minor bit error rate (BER), minimum distance to the sink node, and highest residual energy. Data corruption increases while using a single link for data transmission; therefore, the authors introduced an improved version of EERD called CoEERD. The relay node is used among the forwarder nodes for transferring data from the source to the destination. The forwarder node selection process is the same as the EERD, but data are transferred through the relay node when the threshold values are higher than the BER, improving the data transmission on noisy links. Based on the simulation results, the proposed scheme performed better than the existing routing protocols regarding energy consumption and dead and alive nodes. However, high delays due to the node’s cooperation persist.

The authors in [9] proposed a reliable multipath energy efficient-routing protocol (RMEER). This research aims to extend the network’s lifetime and determine the ideal route for delivering information to the specified destination. The entire network is composed of five distinct and equal layers. The destination node is positioned at the water’s surface, and heavy static transmitters are distributed in the other layers. The multipath routing technique is used for data transferring among the sensor nodes. Multilink nodes are used to enhance the packet delivery ratio (PDR). In such a method, if one node dies early in the middle of data transferring, the selection process automatically bypasses the dead node. Based on simulation results, the proposed routing protocol performs better than other routing schemes regarding energy consumption, throughput, and alive nodes. Selecting the new data forwarding node after dying one another increases the latency and decreases the reliability.

The authors proposed a channel-aware energy-efficient two-hop cooperative routing protocol for underwater acoustic sensor networks in [10]. The multihop technique is used for data gathering and transferring it to the sink node, and the cooperation method is used to improve the network reliability. Data transfers in two phases, the first is data transfer through the forwarder node, and the second is the use of the relay node for data transmission. The maximal ratio combining (MRC) method controls the data transmission when both forwarders receive the information. Additionally, the received signal strength (RSS) method is deployed for finding the distance between the two nodes. The simulation result shows that the proposed protocol performs better in stability and energy consumption. In the proposed scheme, only energy consumption was the focused metric, while the latency and transmission loss were not considered, as the multihop technique consumes more energy.

The delay minimization and battery life problems are addressed via the fuzzy vector technique in [11]. A more advanced version utilizes the fuzzy logic technique (FLT). It generates data and then sends them through a multihop method to the sink, considering the maximum residual power required for data transmission. On top of the node’s position, residual energy is a variable in establishing the optimum forwarder. All of the source’s neighbors receive the data packet when broadcast. Data are sent to the next node through the optimum node selected among all the nodes in the network. To keep the node alive, its residual energy should be as high as possible, and its position should be as low as possible so that it does not drop to the bottom. Based on simulation results, the proposed routing protocol performs better than the existing routing protocols regarding data transfer speed and the most significant number of active nodes. The authors focused on only energy consumption through the fuzzy logic method, while transmission loss and packet delivery ratio were not discussed.

The authors in [12] propose a power-efficient routing protocol (PER) to reduce energy utilization in the network by choosing the optimum nodes for packet forwarding. Sending node selector and forwarding tree pruning mechanism are two parts of the PER. The first module uses a fuzzy logic interface and tree decision to select the next two forwarder nodes based on angle, residual energy, and distance. In addition, tree forwarding is employed to discover the most efficient route for packet distribution. In reality, tree forwarding stops additional packets from spreading in the network. The different ocean factors, such as noise and transmission loss, are not considered in the design time.

Regarding the issues of propagation delay, power constraint, and bandwidth, a new method dubbed “multimedia- and multiband-based adaption layer strategies” is proposed in [13]. Using the proposed routing technique, data are efficiently captured and sent. The sink hierarchically receives information, which is how it should be. This task is broken down into two separate steps to select a procedure. Using the Manhattan method, you can identify which nodes are closest and farthest from your final destination and which media is best for sharing data. Using RSSI, you may estimate how far something is from you. The multimedia modem supports the 70 to 140 kHz bandwidth of acoustic waves. An infrared wavelength range from 700 nm to 1 mm only employs a single bandwidth, and blue light with a wavelength of 450 to 485nm is used to communicate in the visible light spectrum. Simulations show that the suggested method has superior energy consumption, bandwidth, and propagation delay to the current standard.

The main focus in [14] is the network’s stability and noise analysis. These two factors focus on a routing technique known as a depth-based noise-aware scheme. The whole network is entirely unaware of the existence of any nodes at first. The channel reciprocity rule is observed when a hello packet is sent from the source to the destination. Every node receives its ID number, depth, and noise information from this hello packet. As a result, each node gets to know the others. Every node in the source transmission range receives the information signal when it generates it. There is a massive energy crisis if all network nodes submit this info to the sink node. Only one node is chosen to transmit the packet in this case. This node selects the parameters with the lowest depth and noise. Such a sink node is determined to complete the data transmission successfully. This node should be set to reduce energy usage and improve signal quality. Furthermore, the network’s stability period improves as the number of live nodes grows and the number of dead nodes decreases. High network traffic on the relay node issue persists.

The upcoming parameters affect the network’s overall performance in packet delivery, battery life, and data error rate because of the harsh and unpredictable environment in underwater wireless sensor networks. New approaches have been offered in [15] to deal with these problems. This protocol partitioned the network into four sections based on the node depth. The sensor nodes are divided into depth-based zones: the lowest depth node region, the medium depth node region, and the maximum depth node region. Each of these regions has a different depth. A sink node in each area interfaces directly with the ocean sink node. Only one random node from each zone transfers data to the sink node in that region. Each region has its random nodes. It is decided which forwarder node to use based on its low depth value and high residual energy. The network lifetime, throughput, and reliability are all improved due to this configuration. As only one random node is used for data transmission, the chances of data loss will increase.

Hole formation occurs when the energy of a top node runs out, preventing any further data transfer. Hole creation is avoided in [16]. Dispersed energy-efficient and connectivity-aware routing methods are discussed because sink nodes are far from the region of interest. Underwater data transmission is disrupted when commonly utilized overhead lowest depth nodes create a hole in the network. These solutions prevent this problem. Multipath routes are used in this protocol. Low-depth nodes have a lower probability of going dead. Still, a simulation reveals that their lifespan improves from 50% to 70% due to lower-depth nodes bearing less data transmission load. The number of dead nodes decreases while the number of live nodes reaches a maximum.

The authors in [17] proposed a delay-intolerant energy-efficient algorithm known as DIEER. This approach can increase the PDR to avoid collisions in data packet propagation delays. Except for the DIEER protocol, no other routing protocol addresses these issues. As data are retransmitted less frequently, the network uses less energy, which reduces latency. The joint optimization framework for sink mobility, hold, and forwarding mechanisms is introduced. Data aggregation and pattern matching algorithms fix the threshold’s adaptive value and reduce network delay while delivering maximum data, increasing the network’s life, and using the least energy necessary. An underwater platform with sink mobility and extensive distribution of nodes with different communication radii has been designed for three-dimensional operation. No retransmission of data occurs as a result of adapting the offered protocol.

In-band communication where the TX and RX are required to have reliable communication. Through the TX–RX interaction, reliable communication protocols can ensure lead performance. The author of [18] designed and developed the Onion, a dependence-based communication protocol for MIMO MRC-WPT. It is the extension of the C1G2 protocol. The authors designed and developed the prototype for the implementation of the evaluation of this protocol. The experiment results show that the proposed protocol performed 40% compared to other dependency schemes in terms of communication ratio. An overview of all the schemes is illustrated in Table 1.

## 3. Proposed Protocol Design Scheme

The proposed CEER routing protocol is briefly explained in this section. The CEER is a cooperative routing scheme that reduces energy consumption and keeps the network for a long time.

### 3.1. Energy Consumption and Network Model

Three-dimensional (3D) UWSNs comprise equally distributed sensor nodes deployed in surveillance fields. The 3D UWSNs model is represented by G=(V, E) with m number of sensor nodes in this work. Three-way coordinates (x, y, z) are assigned to every node. Moreover, with localization services in [19], we assume that every sensor node has information about its location. Underwater systems with fixed bottom-mounted nodes that already have location data can make such an assumption. Anchor nodes cannot always be deployed at the seafloor for Deepwater environments. An autonomous underwater vehicle (AUV) with a self-driving system is used as a reference node in the distributed localization algorithms in this scenario. Three-dimensional (3D) Euclidean space is defined by the function δ(u, V) as the distance between two points sv and su as follows:(1)δ:N×N→ Γ : δ(u, V)

In UWSNs, every node has sensing devices for data gathering. They gather data from the exterior environment and send it to the sink node over one or multiple hops. The sink node is the node that produces data gathering results and is also the target site of transmitting data. Each sensor node can send and transmit data packets depending on its configuration. Every node can adjust its communication range from *r*_min_ (minimum transmission radius) to *r*_max_ (maximum transmission distance). The Euclidean distance (*u*, *V*) between two sensor nodes is bound when the distance between them is constrained by u(h) ≤ δ(u, V) ≤ V(h). Two sensor nodes have the same minimum hop distance (*h*). Because of this, network density *ρ* has a significant impact on boundary quality. In specific for each case where h>0 obtains
(2)$$lim\;V\left(h\right)-u\left(h\right)=r_{min}$$
where $r_{min}$ shows the lowest communication range of the sensor nodes, a wide range of physical and theoretical characteristics can be seen in sensor networks. Therefore, a wide range of models is created based on the application’s needs and the device’s capabilities [20]. However, in most sensor nodes, sensing ability is reduced when the distance between the sensors and the objects increases. The general model for sensing in underwater wireless sensor networks is shown as:(3)$$d\;\left(s,\;p\right)=\frac{\lambda }{\left[d\;\left(s,\;p\right)\right]k}$$
where d (s, p) the Euclidean distance between *s* and the point *p*, *k* is the sensor parameter, and λ is the positive constant in [20]. We consider that all the sensor nodes have low battery capabilities and are not recharged or replaced after implementation. The network’s lifetime is described as the time it takes for the first sensor node in the network to run out of energy.

The simulation area is divided into four equal squares labeled upper right square (URS), upper left square (ULS), bottom right (B.R.), and bottom left (B.L.). Sink nodes (S.N.) move in a three-cornered path to gather data from the sensor nodes in each portion. Randomly installed nodes sense the characteristics and transform them into packets. So that they can be used in consequent processing steps, the data are sent to the sink node. Each of the nodes has direct communication with the sink nodes.

Transmission mechanisms of acoustic signals differ between shallow and deep water. There are two ways to describe the audio signal transmission in shallow-water and deep-water spherical diffusion, and the energy consumption is caused by spherical distribution with water absorption.

The signal-to-noise ratio of an acoustic signal on the receiver side can be calculated as [21]
(4)SNR=SL−TL−NL+DI

*TL* is the transmission loss (dB), *NL* is the noise level, *SL* is the receiver source, and *DI* is the directivity index. Calculate the transmission loss of circular spread signals, then it is calculated by
(5)$$TL=10log_{2}\delta \left(u,\;V\right)+\alpha \delta \left(u,\;V\right)\times 10^{-3}$$

Here, (*u*, *V*) is the distance between the sender and receiver and is the medium-dependent frequency coefficient in dB. Throop’s empirical formula gives the absorption loss in [22].
(6)$$\alpha =\frac{0.11f^{2}}{1+f^{2}}+\frac{44f^{2}}{4100+f^{2}}+2.75\times 10^{-4}f^{2}+0.003$$

Here, α is in dB/Km and *f* is in KHz.

Waves, shipping, wind, and considerable mammal activity can alter NL’s noise level in shallow water. For simplicity, an average noise level NL in shallow water is taken as 70 decibels [22]. The following statement can be used to express the relationship between the transmitted signal intensity at one meter from the source and the radiated sound intensity in decibels [23]:(7)$$SL=10\;log_{2}\;\frac{I_{t}}{1\;\mu \;Pa}$$

Here, $I_{t}$ is in μ Pa and solving for $I_{t}$ Yields.
(8)$$I_{t}=10^{SL/10}\times 0.67\times 10^{-18}$$

Thus, the $P_{t}$ achieves intensity is the transmitter power. Distance from transmitter to receiver is calculated as 1 m at this point [23].
(9)$$P_{t}=2\pi \times H\times I_{t}$$

Here, H is the depth of water in meters, and $P_{t}$ is in watts.

The function parameters (energy, distance, and bit error rate) are considered to select a destination. The packet arrives directly at the sink node from a neighboring source node. Otherwise, the packet is sent by the source node considering the multihopping. The parameters for selecting a destination where data will most likely be forwarded are listed below.
(10)$$f=\frac{Residual\;Energy}{Distance\times BER}$$

The node with the highest residual energy and the lowest BER is designated as the first destination, according to Equation (10). The best relay node is selected if the BER falls below a certain threshold. The source directs the data packet to its final destination once the destination has been chosen.

### 3.2. Routing Strategies

The proposed routing protocol strategies are divided into two stages: discovering possible candidates and selecting relay nodes. Candidate discovery is shown in Figure 2. A different icon shows each sensor node’s remaining energy at a particular sensor node. With RF and acoustic modems installed at the water’s surface, sink nodes do not need to care about running out of energy. Sinks that do not move about only need to broadcast their locations once during startup, which does not require significant energy. The transmitter sensor nodes are the nodes that hold the data packets. Every data packet carries the location of the sink node, relay node, and source sensor node. Before the data transmission, the si is the sender node, and the other nodes are the receiver nodes, as shown in Figure 2. The hello packets are transmitted by the si node with a minimum radius $r_{min}$ including the position of itself si and the position of the st sink node. The cosine of the direction between the si and sj receiver will be calculated by the sj because it is the closest receiver of the transmitter. The advertisement packet with a radius of $r_{j}$ will be transmitted by the sj if the cosine value is greater than 0. The radius $r_{j}$ will be calculated as:(11)$$r_{j}=MIN\;\left\{\left(1+\frac{\varepsilon_{j}^{res}}{\varepsilon_{j}^{max}}\right)\;.\;r_{\min},\;r_{max}\right\}$$

Here, $\varepsilon_{j}^{max}$ is the highest energy of the sensor node, and the $\varepsilon_{j}^{res}$ is the starting energy of the sensor node. As a result, the range of $r_{j}$ is $r_{min}$ to 2$r_{min}$. Equation (11) shows that in an ideal situation, ($r_{max}$ > 2$r_{min}$), and that the starting energy of sj is complete, sufficient to fill the transmission circle of si. Only an advertisement packet with a radius of $r_{min}$ May reaches the si location in the worst-case scenario, the remaining energy in sj is nearly consumed. Furthermore, the advertisement packet contains data about sj’s location and residual energy.

When sk collects the sj advertisement packet and si Hello packet, it extracts the residual energy and the position from these packets.

The sk will go into sleep mode and will not transmit any packet to save the power because the residual energy of sk is lower than the sj’s. The hello and advertisement packets are also received by sq because its also in the receiver nodes. However, since the cosine of the angle here between routes from si to sq and from si to st is less than zero, sq will choose to go to sleep mode. Thus, sq is located at a preferred destination compared to other receivers. Although within the broadcast range of sj, the receiver sh remains in sleep mode because sh cannot receive the hello packet sent by si. After receiving si’s hello and sj’s advertising packet, the position and residual energy information are extracted from these packets. Upon receiving a courting packet, the sensor node that has gone into sleep mode will immediately wake up and find itself within the communication range.

### 3.3. Cooperation and Relay Node Selection

All receiver sensor nodes received the data broadcast by the source node Source. The source node follows the cooperation strategies. A simple cooperation model shows in Figure 3, where the Source, the source node, transmits the signals to relay nodes Relay1, Relay2, and the Destination. The signal received by Relay1, Relay2, and Destination is developed as [14].
(12)$$Y_{sd}=X_{s}h_{sd}+n_{sd}$$

Here, $X_{s}$ represents the transmitted signal in its original state. The symbols $n_{sd}$ and $h_{sd}$ represent the noise, and channel gain, respectively, from Source to Destination. The symbol represents the output signal at Destination $Y_{sd}$. The signals sent from Source to Relay1 and Relay2 are described as follows:(13)$$Y_{sr1}=X_{s}h_{sr1}+n_{sr1}$$
(14)$$Y_{sr2}=X_{s}h_{sr2}+n_{sr2}$$

Here, $Y_{sr2}$ and $Y_{sr1}$ are the Relay2 and Relay1 signals output, respectively. The $h_{sr1}$ and the $n_{sr1}$ are the channel noise and channel gain from Source to Relay1 over the link. The $h_{sr2}$ and the $n_{sr2}$ are the channel noise and channel gain from Source to Relay2, respectively. The signal from Relay1 and Relay2 at Destination is modeled as:(15)$$Y_{r1d}=\beta Y_{sr1\;}h_{r1d}+n_{r1d}$$
(16)$$Y_{r2d}=\beta Y_{sr2\;}h_{r2d}+n_{r2d}$$

Here, the received signal from Relay1 to Destination is shown by Yr1d and Yr2d respectively. hr1d and nr1d are the channel gain and channel noise from Relay1 to Destination, respectively. hr2d and nr2d are the channel noise and channel gain from Relay2 to Destination. Algorithm 1 explains the proposed CEER routing protocol.
**Algorithm 1.** Shows the proposed CEER routing protocolSi = Sender Nodesrmin = minimum radiusSt = Sink nodeSJ = Cosign B/W Source and DestinationrJ = Advertisement packetSn = All Candidatesp   = routing path**While** (TTL > 0) and (Si ≠ St) doSi.Sn ← Ø;Si Transmit an advertisement packet with a radius rmin**for** all SJ with δ(i , j) < rmin do**if** cos (TTL) < 0 thenSJ.sleep ();**else** St.Sn.add (SJ)SJ transmits an advertisement packet with a radius rJBased on Equation (11);
    **for** all Sk with δ(j , k) < rJ **do**    **if** ( εkres<εjres) then      Sk.sleep();    **else** Si.Sn.add (Sk)    **end if**    **end for** **end if****end for****end while**

## 4. Results and Performance Evaluation of the Proposed and Existing Routing Protocols

The following section defines the simulation scheme of the proposed CEER and compares the CEER and the existing state-of-the-art EELRP and EEDORVA routing protocols. Simulation parameters are illustrated in Table 2. Additionally, this section describes the primary performance metrics for all compared protocols.

### 4.1. Performance Metrics

Performance metrics for all compared protocols are defined as follows:

**Residual Energy:** It describes the difference between the startup nodes’ energy and the energy of the nodes utilized during the operation.

**Network Lifetime:** The total time spent by the operation of the network is referred to as network lifetime.

**Throughput:** The total number of efficiently transmitted packets at the sink is called throughput.

**Path-Loss:** The difference between sending and receiving nodes’ transmitted and received powers is called path-loss and is measured in decibels (dB).

**Channel Loss:** channel attenuation is typically expressed in decibels (dB) per unit distance. Attenuation of zero decibels means that the signal is passed without loss; three decibels means that the power of the signal decreases by one-half.

### 4.2. Results and Discussion

This section describes the results of the proposed routing protocol and the comparison with state-of-the-art protocols such as EERP and EERD-VP. This research work is modeled in a 3D environment with a height of 1000m x 1000m x 1000m using the random-walk mobility pattern, and the sensor nodes move around. Between the minimum and maximum speeds, 0m/s and 3m/s, each sensor node randomly selects a direction and moves to the new location at random speeds. The bit rate is 10 kbps, and the communication settings are identical to those on a commercial acoustic modem. Each data packet’s time-to-live (TTL) value is set to 20.

The sensor nodes with the same physical properties, such as depth and weight threshold, work together to keep each other informed about network conditions changes. Sensors communicate with each other and the higher layer through other sensors till the information reaches the sink node, where it can be used. The sink node is in charge of the depth thresholds and mobility of the sensors that work with it. The CEER method can be used when sink mobility, collaboration, and depth thresholds are added in data-critical situations.

To understand a clear idea about the stability and instability periods of the proposed and current state-of-the-art routing protocols, we have decided to analyze and show the alive sensor nodes in both the proposed and existing protocols. Figure 4 illustrates that CEER is better than both the existing routing protocols. The first or initial node in the CEER dies at 1200 s, and is approximately 200 s longer than the EELRP and EEDORVA. In the EELRP and EEDORVA routing schemes, node dying starts gradually after 1000 s. CEER was found to have a more extended period of stability than the existing techniques since it was stable for much time. Compared to the EELRP, the EEDORVA is better after our proposed protocol. The figure illustrates the network’s lifetime with the proposed protocol has been enhanced by 15% approximately compared to the EEDORVA and more than 20% from the EELRP.

The energy consumption of the proposed and existing EELRP and EEDORVA routing protocols is shown in Figure 5. The total energy consumption of the CEER is lower than the two other schemes due to efficient data forwarding through the sink nodes. Data are forwarded directly when the sink node is in communication range to minimize energy consumption. When the sink is not in communication range, CEER selects the nearest neighbor node with maximum energy and bit error rate (BER) for data transferring to the sink node. The EELRP routing scheme gathers data in multiple steps because the authors divided the total simulation area into various segments. Each segment has its agent for data gathering and then transfers to the final destination. From the start of the simulation, CEER uses low energy compared to the other two schemes and then decreasing gradually when the time increases.

Figure 6 shows the packet delivery ratio (PDR) comparison of the CEER and the other two existing routing protocols. The graph illustrates that the CEER performs better in terms of PDR than the other two schemes. The main reason for the high PDR is focusing on lower BER, and the CEER scheme can achieve high PDR by avoiding adverse channel effects. In addition, it uses essential function variables of highest residual energy and lowest distance to select sender and destination. The CEER strategy decreases the number of participating nodes by working with direct communication between the source and destination, which uses minimal energy and keeps the nodes alive for a long time to transmit the packets. The figure illustrates that the proposed routing scheme enhanced the PDR by 20% approximately compared to the EEDORVA and more than 25% compared to the EELRP.

The end-to-end delay of the proposed and existing routing protocols is plotted in Figure 7. As shown in the figure, the delay of the proposed scheme is lower than all other schemes because of the lower distance between the sensor and sink node. The high delay in EELRP is to transfer data from multiple stages and not directly sent to the base station. As illustrated in the graph, the EEDORVA performs better than the EELRP because the EEDORVA transfer data to the base station using simple depth value selection. The figure demonstrates that the proposed routing scheme enhanced the end-to-end delay by 200% approximately compared to the EELRP and more than 25% compared to the EEDORVA.

The transmission loss of the proposed CEER and the EELRP and EEDORVA schemes is plotted in Figure 8. As illustrated in the graph, using the multiple cooperative nodes is very effective in the CEER compared to the EELRP and EEDORVA noncooperative techniques.

The proposed CEER scheme uses the Thorps attenuation model to measure the total loss between the source and destination during the data transmission. The other two methods, EELRP and EEDORVA, start lower and increase as the rounds increase. The figure illustrates that the proposed routing scheme enhanced the transmission loss by approximately 28%compared to both the existing EELRP and EEDORVA schemes.

## 5. Conclusions

This paper proposed the CEER for UWSNs. The proposed routing protocol significantly improved the PDR, end-to-end delay, energy consumption, stability period, and transmission loss. CEER uses the sink mobility technique for energy consumption, and for the network’s reliability, it uses the cooperative method for data transferring. The simulation results show that the proposed scheme is efficient from the current routing protocols in evaluated performance metrics. The results also indicate that CEER is better up to 25%, 25%, 200%, and 28% in terms of energy consumption, PDR, end-to-end delay, and transmission as compared to the EELRP, respectively, and 25%, 20%, 25%, and 28% as compared to the EEDORVA, respectively.

## Figures and Tables

**Figure 1 sensors-22-06945-f001:**
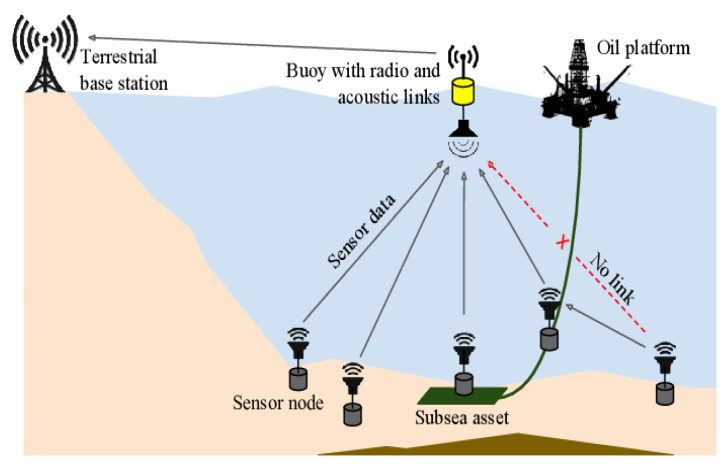
UWSN architecture.

**Figure 2 sensors-22-06945-f002:**
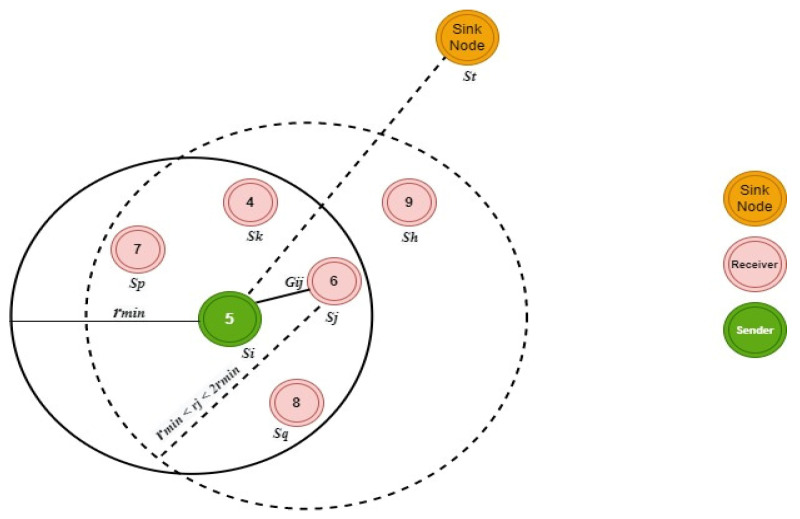
Candidate discovery phase.

**Figure 3 sensors-22-06945-f003:**
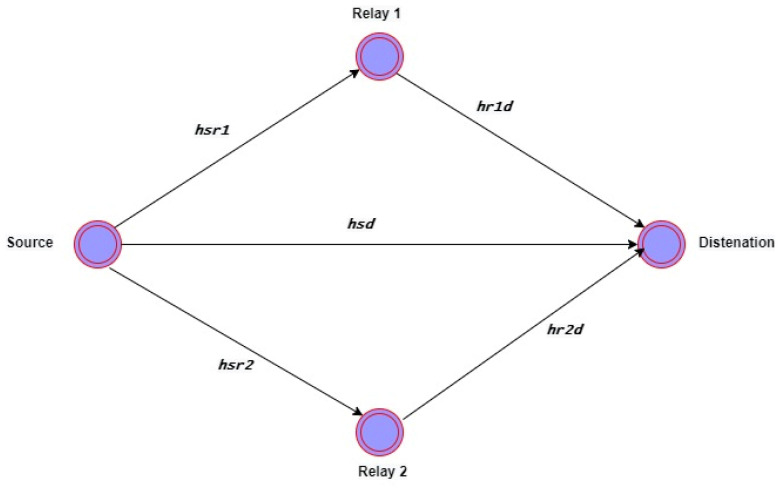
A simple model of cooperation.

**Figure 4 sensors-22-06945-f004:**
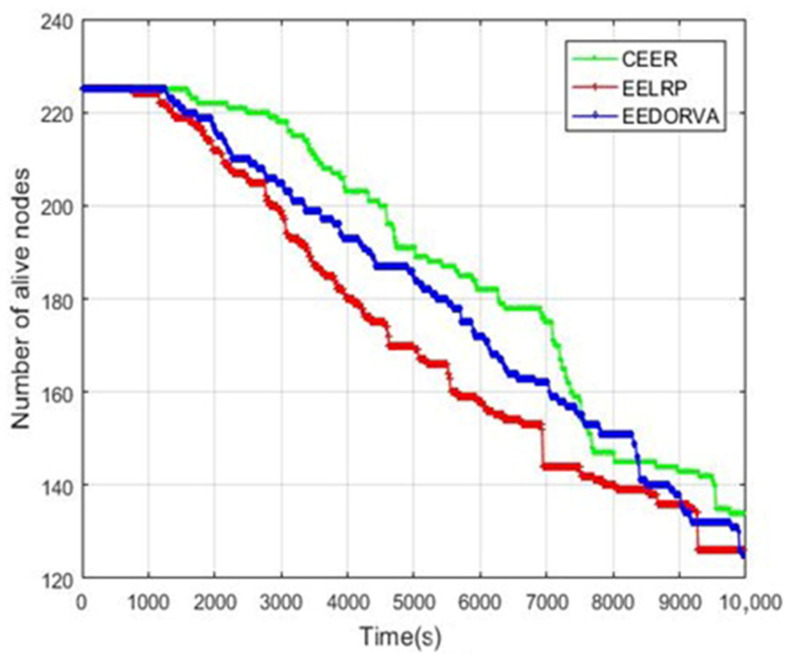
Alive nodes versus of network’s lifetime.

**Figure 5 sensors-22-06945-f005:**
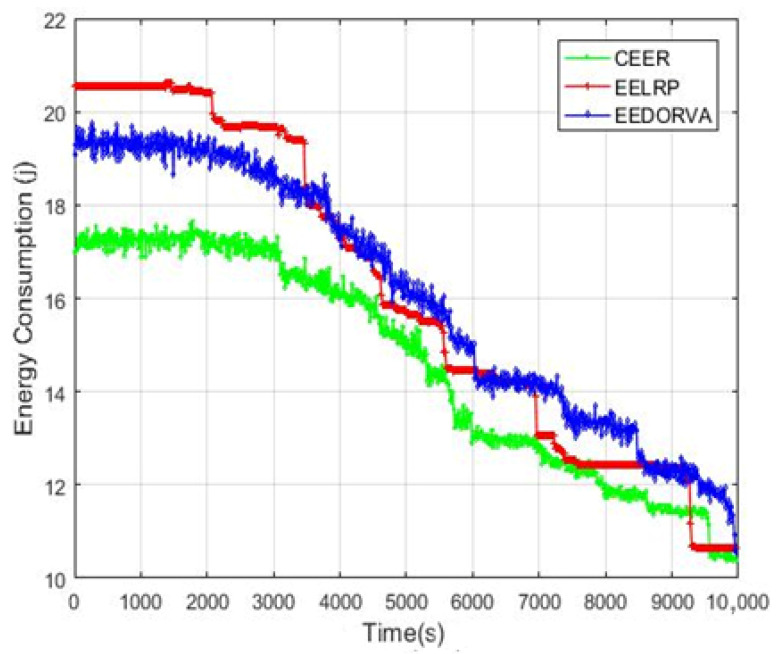
Energy consumption.

**Figure 6 sensors-22-06945-f006:**
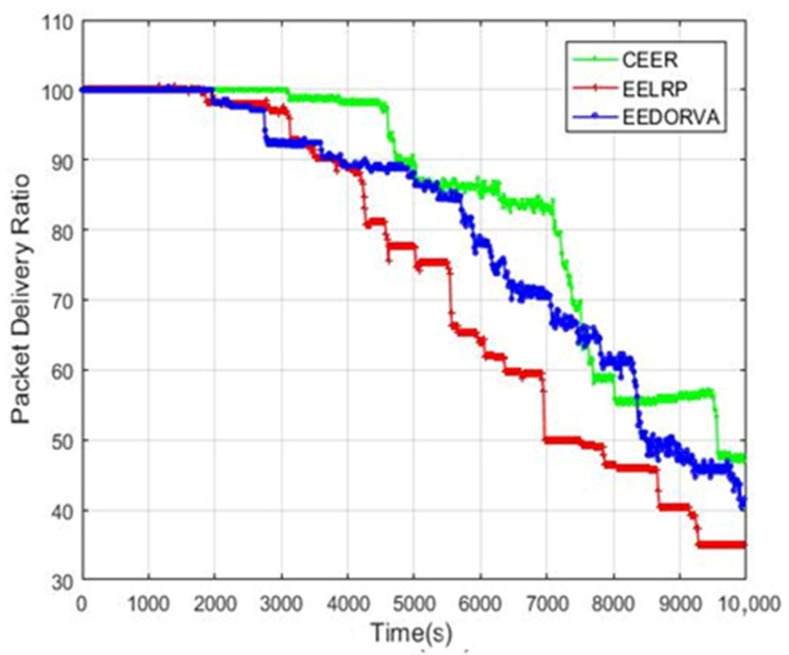
PDR of the routing scheme.

**Figure 7 sensors-22-06945-f007:**
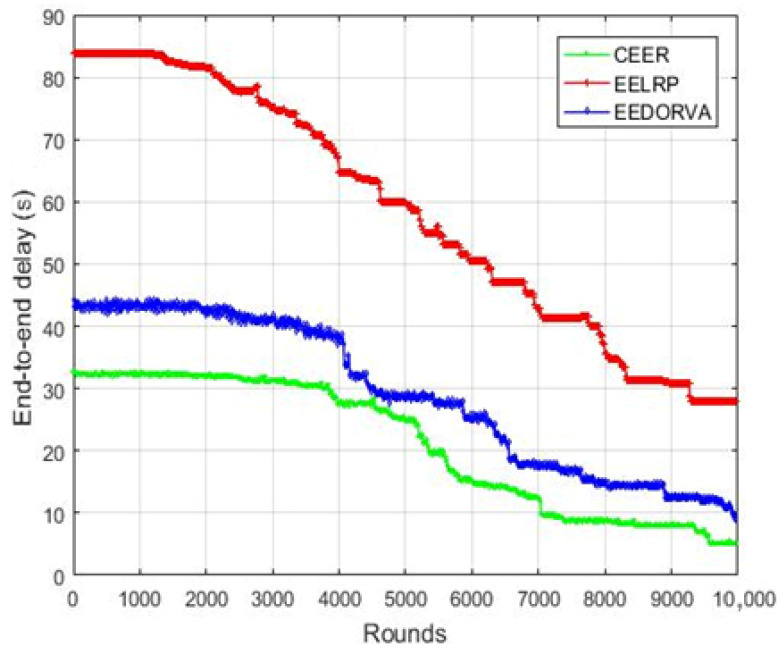
End-to-end delay of the proposed and existing protocols.

**Figure 8 sensors-22-06945-f008:**
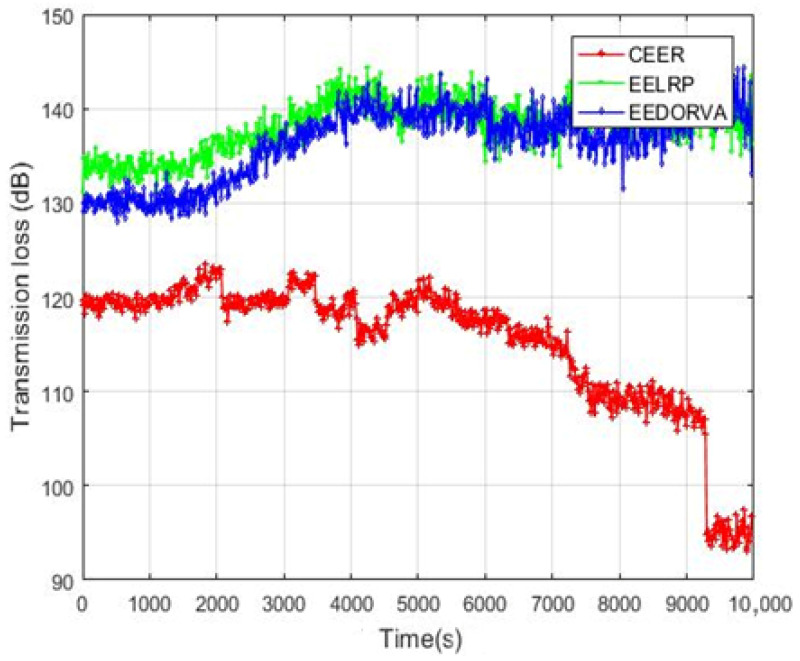
Transmission loss of proposed and EELRP and EEDORVA.

**Table 1 sensors-22-06945-t001:** Overview of the recently published schemes.

Citations	Year	Technique Used	Advantages	Shortcoming
[8]	2019	Single and multipath routing schemes, data forwarder nodes are selected by considering the minor bit error rate (BER), minimum distance to the sink node, and highest residual energy technique.	Improved performance in energy usage and reliable packet transfer.	High delays due to the node’s cooperation.
[9]	2018	The multipath routing technique is used for data transferring among the sensor nodes.	Improved the packet delivery ratio (PDR)	Increases the latency and decreases the reliability.
[10]	2019	Cooperative multirelay scheme, best forwarder nodes are selected by considering the weight function, use MRC technique.	Improved the network reliability	Consumes more energy.
[11]	2018	Fuzzy logic technique (FLT) with multihop method sends data to the sink.	Data transfer speed and the most significant number of active nodes.	Due to multihop high delay and maximum energy consumption
[12]	2011	Fuzzy logic interface and tree decision to select the next two forwarder nodes based on angle, residual energy, and distance.	Improved energy consumption, end-to-end delay, and PDR.	The different ocean factors, such as noise and transmission loss, are not considered.
[13]	2019	Manhattan and RSSI methods are used for data transmission.	Improved propagation delay, power constraint, and bandwidth.	High delay due to hierarchical structure method.
[14]	2019	Both the cooperative and noncooperative schemes used the node position information by considering distance and mobile sinks for information advancement.	Decreases latency and increases the throughput.	Consumes maximum energy due to the deficiency of the balanced energy technique.
[15]	2015	Divided the area into four regions, each region only one random selected node transfers the data to sink.	Improved network lifetime, throughput, and reliability.	Data transferring through only one random node chances of data loss will increase.
[16]	2017	Non-cooperative base routing scheme, used a region of interest.	Improved network lifetime, node loss rate, and network overhead	Less reliability, high propagations delay.
[17]	2020	joint optimization of sink mobility, hold and forward mechanisms, adoptive depth threshold (DTH) and data aggregation with pattern matching	Increase the PDR to avoid collisions in data packet propagation delays.	Consumes more energy

**Table 2 sensors-22-06945-t002:** Simulation parameters.

Parameters	Value
Simulation Deployment Width	500 m
Simulation Deployment Depth	500 m
Simulation Deployment Breadth	500 m
No. of Sensor Nodes	225
No. of Sink Nodes	10
Transmission Range	220 m

## Data Availability

Data that supports its result is available upon request from the first author.

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
