# Peer review of "Cooperative Energy-Efficient Routing Protocol for Underwater Wireless Sensor Networks"

_sensors, 2022, doi:10.3390/s22186945_

Round 1

Reviewer 1 Report (New Reviewer)

This paper aims to investigate the routing issue of underwater wireless sensor networks. The proposed cooperative energy efficient routing (CEER) protocol has significantly improved the PDR, End-to-End delay, energy consumption, stability period and transmission loss. Some results are given in detail, and the authors have made great effort on this paper. However, there are still some problems, which are listed as following.

1.Some existing results are mentioned in the literature review, but the shortage of the existing results and the advantage of this paper are not given, so it is hard to find some innovation and merit comparing with other literature. Could the authors give brief introduction about the shortcomings of them? Or emphasize the out performance of this paper.

2. How to select the relay node in Section 3.3, I don’t understand. What is the relationship between Section 3.3 and CEER routing protocol.

3. Several misprints need to be corrected. For example, in line 235 of page 6, “where r_{min} Shows …”; in line 252 of page 6, “… to the Sink node”; in lines 303-304 of page 7, “The Hello packets are transmitted by the si node with a minimum radius r_{min} Including the position of itself si and the position of the st the sink node”.

4. Some parameters used in simulation need to be given. For example, $beta$ in Equations (13) and (14).

5. Double-checking citations, e.g., [1],[2],[5],[8],[9],[10],[12],[19],[23].

6. Presentation aspects need more care. Several figures are too vague.

7. The description of this paper is very confusing. The authors should reorganize this paper to make it clearer and more understandable.

Author Response

Dear Reviewer,

Thank you for your time to check our Manuscript and pointed out some points which have been incorporated into the revised Manuscript.

A point-by-point response to reviewers can be seen in the attachment.

Regards,

Authors

Reviewer 2 Report (New Reviewer)

This paper deals with a hot area of investigation at the moment. The work is well organized and appropriately carried out.  I have looked at the mathematics and it looks sound.  

*In Section 1 (Introduction), the presentation of this section is poor. Suggest dividing this section as 4 subsections (1.1. Background, 1.2. Motivations, 1.3. Contributions, and 1.4 Paper organization)

*In section 2, the study lacks a clear comparison between the submitted paper and the more relevant literature contributions, which should highlight the main advantages of the current submission. So, Suggest summarizing Section 2 in the Table at the end of Section 2 and support with more recent publications.

*Section 2 did not discuss MIMO-WPT?

Author Response

Dear Reviewer,

Thank you for your time to check our Manuscript and pointed out some points which have been incorporated into the revised Manuscript.

A point-by-point response to reviewers can be seen in the attachment.

Regards,

Authors

Round 2

Reviewer 1 Report (New Reviewer)

Thank you for answering the questions, and revising the article. Paper can be recommended for publication.

Reviewer 2 Report (New Reviewer)

The authors have addressed the comments.

This manuscript is a resubmission of an earlier submission. The following is a list of the peer review reports and author responses from that submission.

Round 1

Reviewer 1 Report

The paper addresses the Underwater Wireless Sensor Networks (UWSNs) with the usual challenge of powering sensor units. Hence, the authors are exploring a mean to reduce the UWSN power consumption. One of the critical energies draining is the routing process. Their aim is to elongate the network lifetime, before the need of near impossible process of replacing batteries, while maintaining the network reliability. The paper focus to alleviate the energy consumption issue is the UWSN via the introduction of  “Cooperative Energy Efficient Routing Protocol (CEER)” to increase the network lifetime and reliability of the network.

Good related section chapter, scientific approach and network design analysis

BUT:

Please think one million times of rewriting your abstract, it is a very naïve paragraph that does not at all reflect the significance of your work and your efforts in the paper. My advice is to read carefully your introduction and then write your abstract again, identifying the significance of the research point/problem in the critical military/civil applications/domains, spelling out the true design of your network (mobile sink or-and nodes). Please make sure of your technical writing of your abstract as it is the first stop of any reader of our respected journal! For example, line 20 (also similar is repeated in line 25) “… hot area for industrial and researchers…” a very odd and non-technical.

63-64   “….designed well and 63 managed;” technical writing (TW) à “….well designed and managed;” is much better.

76         “…Cooperation transfers data…” , I think you mean “…Cooperative communication transfers ….”

87-88   “However, these techniques are unreliable and have a higher probability of…… data packets.” I am missing a word like
             “dropping”.

88-89   “A minor key obstacle can result in data loss and 88 failures.” Vague What is the “minor key”? I think it is TW

92-93  “Cooperation is the optimal strategy for ensuring reliable and effective communication in UAWSNs.” It is a big claim that
               requires a citation

95        “….The authors used the void avoidance technique..” a bit odd (TW)

104-105 “..The energy of sensor nodes and data transferring in the harsh environment through noisy fewer links are the two   
                      most important issues are discussed in..”, another TW problem

112-113   “..There are many chances of data corruption while using a single link for data transmission;..”, another TW, choice of
                  wording.

149-150  “…Based on our findings, we can confidently 149 say that our network has the best answers in terms of data transfer
                  speed and the most 150 significant number of active nodes…
” I think this is out of context, I am not sure such
                  comparison relates to which work [10], or [9], or the future down below [11] [12][13] etc? You should make your
                  statement about your work individually at the end of each peer work, or not in the middle, or cumulatively at the end of
                 the chapter.

214         “…protocol offered….” à “….offered protocol ….”.

230         “?:? × ?→ ?∶ ?(?,?……..(1)”  Please, explain the equation involved terms, no need to wait for later and guess too.

245         “……Therefore, a wide range of models is created based on the application's needs and 245 the device's capabilities…”
                    you need to list citations to support your claim

303            “..Every data packet transmits the 303 location of the sink node, relay node, and source sensor node…”, a packet does not
                   transmit
? I think you mean “carries” not “transmits”.

307             “The source 337 node follows the cooperation strategies in emergencies.”  What about non-emergencies?

339-340        “…in figure 3, where S, the source node, transmits the signals to relay nodes R1, R1, and D destination” there are no
                        R1, R2, or D in fig 3  Be consistence, either spell it out full words on both, or abbreviate in both

356                “Here, the received signal from R1 to D are shown by ??1? and ??2? respectively” I do not see ??1? and ??2? in the
                       figure, instead I see hr1d & hr2d

416             “…with multiple mobile sink nodes on the water surface….” “……Using the random-walk mobility    
                            pattern, the sensor nodes move around. Between the minimum and maximum 418 speeds, 0m/s and
                      3m/s, each sensor node randomly selects a direction and moves 419 to the new location at random speeds….”

                     I am really confused you started with mobile “sink” nodes, then suddenly simulation UMWSN, M for mobile, which is
                      a completely different game. Please explain. Are you designing UWSN OR UMWSN
?

Please justify normal node mobility and any effect on the routing protocol.

Reviewer 2 Report

The paper proposes a new routing protocol for underwater wireless sensor networks. The topic is interesting, and, as the authors claim, an active area of research. However, I cannot recommend the paper for publication in its current form for the following reasons.

The description of the protocol in Section 3 is too confusing. Some equations, like (3), for example, make no sense nor are they sufficiently explained. I would like the authors to add a brief explanation of the workings and main ideas of the protocol before entering into the specifics of the loss model and destinations. Moreover, please, make the distinctions between destinations, relays, more clear. In Section 3.2, what is a candidate? Is it a destination? A relay? Also... what is the cosine angle used for?

All in all, I would recommend rewriting the whole article adding clarity and order to the explanation before sending it for publication. In its current form, I am not able to judge its technical quality.